# A Comparative Analysis of the Stomach, Gut, and Lung Microbiomes in *Rattus norvegicus*

**DOI:** 10.3390/microorganisms11092359

**Published:** 2023-09-21

**Authors:** Taif Shah, Yuhan Wang, Yixuan Wang, Qian Li, Jiuxuan Zhou, Yutong Hou, Binghui Wang, Xueshan Xia

**Affiliations:** 1Faculty of Life Science and Technology, Kunming University of Science and Technology, Kunming 650500, China; taifshah@yahoo.com (T.S.);; 2Department of Biodiversity Conservation, Southwest Forestry University, Kunming 650500, China; 3Research Institute of Forest Protection, Yunnan Academy of Forestry and Grassland, Kunming 650500, China; 4School of Public Health, Kunming Medical University, Kunming 650500, China

**Keywords:** *Rattus norvegicus*, microbiome, stomach, gut, lung

## Abstract

Urban rats serve as reservoirs for several zoonotic pathogens that seriously endanger public health, destroy stored food, and damage infrastructure due to their close interaction with humans and domestic animals. Here, we characterize the core microbiomes of *R. norvegicus’s* stomach, gut, and lung using 16S rRNA next-generation Illumina HiSeq sequencing. The USEARCH software (v11) assigned the dataset to operational taxonomic units (OTUs). The alpha diversity index was calculated using QIIME1, while the beta diversity index was determined using the Bray–Curtis and Euclidean distances between groups. Principal component analyses visualized variation across samples based on the OTU information using the R package. Linear discriminant analysis, effect sizes (LEfSe), and phylogenetic investigation were used to identify differentially abundant taxa among groups. We reported an abundance of microbiota in the stomach, and they shared some of them with the gut and lung microbiota. A close look at the microbial family level reveals abundant Lactobacillaceae and Bifidobacteriaceae in the stomach, whereas Lactobacillaceae and Erysipelotrichaceae were more abundant in the gut; in contrast, Alcaligenaceae were abundant in the lungs. At the species level, some beneficial bacteria, particularly *Lactobacillus reuteri* and *Lactobacillus johnsonii*, and some potential pathogens, such as *Bordetella hinzii*, *Streptococcus parauberis*, *Porphyromonas pogonae*, *Clostridium perfringens*, etc., were identified in stomach, gut, and lung samples. Moreover, the alpha and beta diversity indexes revealed significant differences between the groups. Further analysis revealed abundant differential taxonomic biomarkers, i.e., increased Prevotellaceae and Clostridia in the lungs, whereas Campylobacteria and Lachnospirales were richest in the stomachs. In conclusion, we identified many beneficial, opportunistic, and highly pathogenic bacteria, confirming the importance of urban rats for public health. This study recommends a routine survey program to monitor rodent distribution and the pathogens they carry and transmit to humans and other domestic mammals.

## 1. Introduction

The mammalian gastrointestinal (GI) tract is densely populated by microorganisms, with their number being approximately 10 times greater than that of mammalian somatic cells. Bacteria, archaea, yeasts, fungi, and protozoa are predominant parts of the microbiota in mammals that are essential for host homeostasis, metabolism, physiology, energy production, digestion, and immunity regulation [1]. Recent studies have revealed that the lung is home to various interacting microbes that regulate immunity and homeostasis [2,3]. Laboratory rats and mice are regarded as the best animal models for biological research and microbial studies due to their close anatomical and physiological similarities to humans [4], but there have been very few studies on urban *Rattus norvegicus* in China [5,6].

*Rattus norvegicus* is one of the most widespread urban pest species, inhabiting many cities worldwide. *Rattus norvegicus* poses a risk to public health because they serve as reservoirs for a variety of pathogenic zoonotic bacteria, such as *Bacillus anthracis*, Yersinia pestis, *Vibrio cholerae*, etc. [7,8], destroy stored food, and damage infrastructure due to their close interaction with humans and other mammals. Few researchers have focused on the comparative analysis of urban rat gut microbiomes in Hainan Province [5,8] compared to those investigating arthropod microbiota. A study revealed 30 phyla in feces from different rat species, including *Rattus norvegicus* (*R. norvegicus*), with Firmicutes being the most abundant, followed by Bacteroidetes and Proteobacteria. In addition, Lactobacillaceae was the richest of the 175 families, followed by Ruminococcaceae and Lachnospiraceae. Furthermore, among the 498 genera, Lactobacillus was the most abundant, followed by Clostridiales and Romboutsia [5].

The role of microbiota in the gut–lung axis [2] has been studied as a two-way phenomenon facilitated primarily by microbiota. The gut–lung axis and immune interactions have an additional impact on the host’s health [3]. Moreover, crosstalk between lung and gut microbiota and secondary metabolites derived from them, such as short-chain fatty acids (SCFA), may have local or distant effects on the host’s health [9]. For example, metabolites that translocate from lung and gut mucosal sites have been shown to regulate the host’s immunity, digestion, and other physiological functions [10,11]. In contrast, changes in the lung or gut microbial diversity (also known as dysbiosis) influence the progression of common respiratory disorders, including asthma and obstructive pulmonary disease, in rodents and humans [9,12]. Intestinal dysbiosis may also influence host-related functions and increase susceptibility to obesity, inflammatory bowel disease, and autoimmune disorders [13,14]. According to data, gut dysbiosis along the gut–liver axis also stimulates innate immunity, pathogenesis, and the emergence of chronic liver diseases [15]. Inflammation and loss of intestinal integrity have also been linked to gut dysbiosis and the transfer of microbiota-associated molecular markers like cytokines and lipopolysaccharides through the portal vein into the liver [16].

The genus Rattus consists of over 60 known rodent species globally [17], particularly in Asia, with various species diverging in different parts of China [5,8]. Four species in this genus, i.e., *Rattus tanezumi* (Asian house rat), *Rattus rattus* (black rat), *R. norvegicus* (brown rat), and *Rattus exulans* (Pacific rat), have been widely distributed in China [5,6]. *R. norvegicus* are social animals that began to live in colonies near farms as agriculture developed in China due to the availability of consistent food sources. Later on, this rodent species spread gradually from China to Southeast Asia, Japan, Russia, and Europe. Given their omnipresence and close association with humans, it is vital to understand the gut, lung, and stomach microbial composition of *R. norvegicus*, collected from Yunnan Province, China, which will be of great interest.

This study aimed to investigate the stomach, gut, and lung microbial communities of *R. norvegicus* collected from the Panlong District, Yunnan, China, using 16S rRNA gene-sequencing analysis. We also explored beneficial, opportunistic, and highly pathogenic bacteria in the lung, stomach, and gut of *R. norvegicus*.

## 2. Materials and Methods

### 2.1. Study Site and Sample Collection

A total of seven adult healthy *R. norvegicus* rats (247.33 g ± 51.25) were captured using live animal cage traps between May and August 2022 in Panlong District, Kunming, Yunnan Province, China, in an area dominated by vegetation and grass, with traps set before sunset and checked in the early morning. All the trapped rats were identified with a unique ear tag.

The experiment was completed in compliance with the guidelines for the care and use of laboratory animals, Faculty of Life Science and Technology, Kunming University of Science and Technology, Yunnan, China (protocol no. 16048).

All the captured rats were kept in a specific pathogen-free environment. Stomach, gut, and lung samples were collected from the rats under sterile conditions. For the stomach (*n* = 7), gut (*n* = 7), and lung (*n* = 7) sample collection, the rats were euthanized by Fatal-Plus intraperitoneal injection, and the chest fur was first washed with alcohol before being washed with 0.9% cold normal saline. All 21 collected samples were washed with sterile normal saline (0.9%), immediately frozen in liquid nitrogen, and stored at −80 °C until DNA extraction.

### 2.2. DNA Extraction and Bacterial 16S rRNA Gene Amplification

Genomic DNA was extracted from stomach, gut, and lung samples using a TIANamp Genomic DNA Kit (TIANGEN Biotech., Beijing, China) according to the manufacturer’s instructions. The hypervariable regions (V3–V4) of the 16S rRNA genes were amplified with PCR using gene-specific primers, i.e., forward 338F: 5′ACTCCTACGGGAGGCAGCA3′ and reverse 806R: 5′GGACTACHVGGGTWTCTAAT3′. Approximately 100 ng of gDNA template, Taq DNA polymerase (Invitrogen, Life Techn., Carlsbad, CA, USA), and a gene-specific primer were used for PCR amplification. The conditions set for the PCR machine (Thermal Cycler, ThermoFisher Sci., Waltham, MA, USA) were initial denaturation at 94 °C for 3 min, followed by 32 cycles (94 °C for 30 s, 55 °C for 30 s, and 72 °C for 30 s), and final amplification at 72 °C for 7 min.

### 2.3. Library Construction and Sequencing

The amplified PCR product (470 bp) was purified for library construction using a QIA-quick PCR purification kit (QIAGEN, Hilden, Germany) following manufacturer protocol. Sequencing libraries were built with an Ion Plus Fragment Library Kit (Thermo Fisher Sci., Waltham, MA, USA). Library quality was assessed using a Qubit 2.0 fluorimeter (ThermoFisher Sci., Waltham, MA, USA) before sequencing on an Illumina HiSeq platform (Illumina Inc., San Diego, CA, USA), following manufacturer protocol. The Illumina analysis pipeline (v2.6) was used to estimate base calling, error, and figure analysis.

### 2.4. Bioinformatics Analysis

The raw sequence data were cleaned by removing reads shorter than 230 bp, primer dimers, low-quality sequences (with ambiguous bases), and Q scores of ≥Q20 using the Illumina quality control toolkit. Primer dimers removed from the library reduce Illumina system specificity for sequencing the desired V4 region of the 16S rRNA gene. The Q30 sequencing score is a standard for high-quality sequencing with virtually no ambiguities or errors. Flash (v1.20) and Pear (v0.9.6) tools were used to merge clean reads into tags with a minimum 10 bp overlap and a 0.1 ‘unoise3′ algorithm from the USEARCH (v11) amplicon analysis tool. The SILVA database (v128) removed chimeras (sequences formed by incorrectly joining two or more sequences) in 16S rRNA data. All the clean sequences were clustered into operational taxonomic units (OTUs) using the USEARCH tool (v11) at 97% homology. Rarefaction curves were created to assess microbial richness based on the OUT data. Such a curve also shows how accurately a specific sample was sequenced to represent its identity. The SILVA database (v128) was used to assign all sequences to different taxonomic groups.

The alpha diversity index (i.e., observed species, Chao1, Simpson, and phylogenetic distance tree) was calculated using QIIME1 (v1.9.1). The beta diversity index was evaluated using non-Euclidean (Bray–Curtis) and Euclidean distance dissimilarities between groups. Clustering and principal component analyses (PCA) were used to visualize variation across samples using the R package (v4.1.2). To identify differentially abundant taxa among stomach, gut, and lung groups, taxonomic biomarker analysis was performed using phylogenetic and linear discriminant analysis (LDA), as well as effect sizes (LEfSe) (v1.1.01). Taxa with a relative abundance (*p* < 0.05) and a LEfSe analysis score greater than 3.0 were designated significant biomarkers. Moreover, random forest tests detected important OTU biomarkers between the groups.

## 3. Results

### 3.1. R. norvegicus Stomach, Gut, and Lung Microbial Compositions

Illumina pair-end sequencing produced 2,625,810 raw reads across 21 rat tissue samples. After assembly and quality filtering, 2,339,489 cleaned tags were obtained for downstream analyses (Appendix A). Each sample received an average of 111,404 reads (range: 103,357–119,215). The Chao1 diversity curves plateaued all samples at good sequencing depth, whereas the Simpson curves failed to level off for the samples (Appendix A), indicating that the analysis has captured most of the bacterial diversities, although more phenotypes can be identified with greater sequencing depth. Clean tags from each sample were aggregated and filtered to generate operational taxonomic units (OTUs) to identify bacterial compositions. Clustering at 97% identity yielded a total of 7094 OTUs and an average of 337.8 OTUs for each sample (range: 109–559) (Appendix A).

To examine the differences in stomach, gut, and lung microbiomes of urban *R. norvegicus*, a total of 21 samples that passed the sequence quality-filtering process generated 301 OTUs in the gut, 804 OTUs in the lungs, and 701 OTUs in the stomach groups (Table 1). The Venn diagram shows 202 OUTs unique to the lungs, 48 to the gut, and 216 to the stomach groups, whereas 395 OTUs were shared among the three groups (Figure 1a). Most of the taxa found in the lung also exist in the stomach and gut. Thus, the 202 OTUs of the lung comprise distinct microbial communities that differ from those of the gut. A highly pathogenic bacterium, *E. coli* (OTU_12), was also found among the stomach-, gut-, and lung-shared groups (Appendix A). The details about the top 15 OUTs among the stomach, gut, and lung groups are shown in Figure 1b.

### 3.2. Comparing the Stomach, Gut, and Lung Microbiomes of R. norvegicus

Distinct microbial compositions were observed between the stomach, gut, and lung groups (Figure 1c). Microbial phyla, such as Firmicutes, Campilobacterota, and Actinobacteriota, were richest in the stomach; Firmicutes, Proteobacteria, and Campilobacterota were higher in the gut, whereas Firmicutes, Proteobacteria, and Bacteroidota were abundant in the lungs. Surprisingly, a low abundance of the anaerobic, Gram-positive bacterium *Turicibacter* sp. LA61, was identified among the stomach and lung groups. The abundance of the two most abundant microbial species, beneficial *Lactobacillus intestinalis*, was reversed from the stomach to the gut, whereas a zoonotic pathogen, *Bordetella hinzii* (*B. hinzii*), was most abundant in the lungs, followed by the gut. Firmicutes abundance gradually increased from the stomach to the gut, while proteobacteria relative abundance was increased in the lung samples. The microbial diversity in the three groups was obvious at lower taxonomical levels. A closer examination of the stomach microbiomes reveals that the Lactobacillaceae, Bifidobacteriaceae, and Helicobacteraceae families were more prevalent. In contrast, Lactobacillaceae, Erysipelotrichaceae, and Helicobacteraceae were abundant in the gut. In contrast, Alcaligenaceae were abundant in the lung group. The relative abundances of different taxonomic levels among the top 20 OTUs in the stomach, gut, and lung groups are shown in Figure 1c, Appendix A.

### 3.3. R. norvegicus Stomach, Gut, and Lung Microbial Diversity

Microbial diversity analysis revealed distinctive microbial communities in the *R. norvegicus* stomach, gut, and lung samples. Alpha diversity quantifies microbial differences between groups. The Chao1 (*p* = 0.009) and Simpson (*p* = 0.02) diversity indexes showed significant differences between the stomach, gut, and lung groups (Figure 2a,b). The Chao1 diversity index showed that the intestinal microbial communities tended to have lower alpha diversity than the stomach and lung groups. At the same time, increased species richness was observed in the stomach compared to the lung group.

### 3.4. Variation of R. norvegicus Stomach, Gut, and Lung Microbial Communities

The Bray–Curtis and Euclidean distance metrics revealed significant differences among the microbial communities in the stomach, gut, and lungs of *R. norvegicus*. Using the Bray–Curtis distance metric, two principal coordinate analysis (PCoA) plots visualized the percentage variations between PCoA1 (25.1%) and PCoA2 (19.2%) among different samples (Figure 3a). A slightly tight cluster was observed between the microbial beta diversity for all groups (stomach, gut, and lung). In addition, two PCoA coordinates based on Euclidean distances revealed percentage variations for PCoA1 (24.1%) and PCoA2 (18.1%) among the samples (Figure 3b), indicating that the visualized percentage variation may be attributed to differences in the physiological conditions of different organs.

Heatmaps based on the Bray–Curtis and Euclidean distance metrics revealed microbial community diversity and similarity among the stomach, gut, and lung groups. According to Bray–Curtis distances, the most similarity and lowest difference in the bacterial community profile was 0.2182, observed between H04 and A02 (gut samples), as shown with yellow triangles (Figure 3c), followed by 0.328 between A04 (gut sample) and D01 (stomach sample), and 0.348 between Z06 and Z05 (lung samples). The detailed values of the heatmap based on the Bray–Curtis distance are shown in Appendix A. The lowest difference was 0.286, observed between A04 (gut sample) and E03 (stomach sample), as shown with orange triangles (Figure 3d), followed by 0.303 between A02 and H04 (gut samples), suggesting that the bacterial profile in these samples was evolutionarily more similar. The detailed values of the heatmap based on the Euclidean distance are shown in Appendix A.

### 3.5. Evaluating Unique OTUs in Lungs

The stomach, gut, and lung groups shared 395 OTUs (Figure 3c), accounting for approximately 30.91% (395/1278) of the total OTUs. However, 202 OTUs were only found in the lungs, implying a distinct lung microbial community. We then used the R package to extract 202 OTUs only present in the lung. We identified abundant environmental-associated bacteria, including Acinetobacter (OTU_4503, OTU_2974, and OTU_213), Bacillus (OTU_1493), Campylobacter (OTU_2643), Corynebacterium (OTU_1502), etc. Moreover, the proportion of butyrate-producing bacteria, Lachnospiraceae (OTU_4488, OTU_2087, OTU_1268, OTU_3887, OTU_3544, OTU_2277, OTU_1142, OTU_1432, and OTU_3597), and Ruminococcaceae (OTU_3273, OTU_1343, OTU_940, OTU_1503, and OTU_1183), was also detected in the lung group (Appendix A). In brief, *R. norvegicus* may contribute to variation in microbial communities in the stomach, gut, and lung OTUs, but not in the lung only.

### 3.6. Evaluating Beneficial, Opportunistic, and Highly Pathogenic Bacteria at the Species Level

A closer inspection of the stomach, gut, and lung microbiomes at the species level revealed several beneficial, opportunistic, and highly pathogenic bacteria. Pathogenic bacteria identified in the lung samples were *Corynebacterium pseudodiphtheriticum*, *Streptococcus parauberis*, *Rickettsiella agriotidis*, *Empedobacter brevis*, *Porphyromonas pogonae*, *Acetobacterium wieringae*, *Rahnella aquatilis*, etc. (Appendix A). In the stomach, we identified *Corynebacterium pseudodiphtheriticum*, *Neisseria zaloph*, and *Rhodococcus corynebacterioides* (Appendix A), whereas in the gut, we identified *Helicobacter muridarum* (Appendix A). The 395 shared OTUs comprise more bacterial pathogens, such as *E. coli*, *Macrococcus goetzii*, *Limosilactobacillus reuteri*, *B. hinzii*, *Lactobacillus (L) intestinalis*, *Helicobacter muridarum*, *Ralstonia insidiosa*, *Limosilactobacillus mucosae*, *Clostridium (C) perfringens*, *C. disporicum*, *C. paraputrificum*, *C. subterminale*, and *Pseudomonas mendocina* (Appendix A). However, these pathogens need further investigation to confirm their virulence and underlying pathogenesis.

### 3.7. Identification of Potential Microbial OTU Biomarkers

Multivariate statistical analysis, LEfSe (*p* < 0.05; LDA score > 3.0), analyzed differentially abundant taxonomic biomarkers of the gut (red), lung (green), and stomach (blue) samples. The relative abundances of 156 taxonomic biomarkers were significantly higher in the lungs, followed by 50 in the stomach and 27 in the gut. Specifically, Bacteroidales, Actinobacteria, Prevotellaceae (SCFA producers that have anti-inflammatory effects), Clostridia, and Bifidobacteriales were abundant in the lungs. Lactobacillus, Streptococcus, Campylobacteria, Lachnospirales, Coriobacteriia, and Blautia were abundant in the stomach group (Appendix A), indicating that these bacteria might play a role in host hemostasis. In addition, the cladogram revealed 68 relatively abundant taxonomic clades, of which 15 clades were in the stomach, 2 clades were in the gut, and 51 clades were in the lungs (Figure 4), which was consistent with the LEfSe result. Moreover, the random forest test determined some important OTU biomarkers with the highest discriminatory power between the three groups (Figure 5). The details about the top 15 OTU biomarkers are shown in Appendix A.

## 4. Discussion

Symbiotic microbes co-exist with urban and captive mammals’ GI tracts and are critical in maintaining host metabolism, physiology, and homeostasis. Unlike the small intestine, where enzymes complete digestion, the large intestine plays no role in digestion. In mice, the GI tract is complex and lengthy, with distinct physicochemical features that structure the resident microbiota [1,18]. Investigators mostly concentrate on feces as a substitute for revealing gut microbiota due to their ease of sampling; however, the mammalian GI tract contains distinct sites with different microbial community compositions [19]. Little work has been conducted to characterize the microbiota outside a model placental mammal. Therefore, we characterized and compared the microbiomes of free-living urban *R. norvegicus’s* lungs, gut, and stomach.

This study explored beneficial, opportunistic, and highly pathogenic microbes in the stomach, gut, and lungs of *R. norvegicus*. Overall, we observed 301 OTUs in the gut, 804 in the lungs, and 701 in the stomach groups. Comparing the microbial compositions of these three groups revealed 395 shared OTUs. The lung microbiome reveals distinct microbial communities and shares most of its taxa with those in the stomach and gut groups. A study reported 3235 OTUs among the feces collected from 34 captive and urban-born rats in Beijing, China [20]. Although we had data for 21 samples from seven free-living *R. norvegicus*, the findings showed highly dissimilar microbial communities along the digestive tract and lungs [20]. Future research will seek to understand the roles of microbiomes in host health and disease.

The gut microbiota includes probiotics (mainly Lactobacillus and Bifidobacteria) and opportunistic pathogens (members of the Escherichia/Shigella, Streptococcus, etc.) [21]. Lactobacillus collaborates with other endogenous, beneficial bacteria to maintain gut homeostasis. Through competition and the production of antimicrobial secondary metabolites, these microbes also play an important role in preventing pathogen colonization [22]. Lactobacillus secretion (lactic acid) is converted to butyric acid by other bacteria in the gut, preventing lactic acid accumulation [23]. Lactobacillus has also been shown to maintain intestinal barrier integrity by acting on epithelial cells. Lactobacillus also stimulates adaptive immune responses by activating macrophages and lymphocytes, indicating its role in the host’s immune response [24]. In addition, Lactobacillus has been shown to bind to mucosal epithelial cells easily and to play an immunomodulatory role by lowering the expression of proinflammatory interleukin and toll-like receptors [25]. Lactobacillus, which is normally beneficial in healthy people, can be invasive and fatal in people with gut dysbiosis [26]. For example, Lactobacillus can cause severe infection in immunocompromised people [27]. Over 200 Lactobacillus-infected cases with various disorders have been reported [28]. Additional probiotic formulations containing Lactobacillus were linked to increased mortality in pancreatitis patients [29].

Ruminococcaceae, Bacteroidales-S24-7, Lachnospiraceae, and Lactobacillaceae are butyrate-producing bacteria that contribute to maintaining intestinal health by regulating metabolism, intestinal microflora balance, intestinal integrity, and anti-inflammatory and immunomodulatory functions in the human gut. A decreased Lachnospiraceae abundance promotes tumor cell proliferation [30], whereas an increase in butyrate-producing Lactobacillus in the lungs was positively correlated with the severity of gastric cancer [21]. An increase in Lactobacillus levels due to inflammation may promote the growth of pathogenic bacteria. In a study, increased lactobacilli abundance in the stomach and gut [31] suggested a stomach–gut axis for gastric cancer development. In another similar study, an altered Lactobacillus proportion in mice with gastric cancer was involved in gastric cancer progression [20]. In addition to Lactobacillus, butyrate-producing bacteria such as *Butyrivibrio*, *Lachnospira*, and *Roseburia* [32] as well as *C. difficile* infection alter immune regulation apart from creating favorable conditions for the invading pathogens. Moreover, abundant butyrate-producing Bacteroidales were found among the lungs, stomachs, and gut, indicating that Bacteroidales in mice suffering from chronic non-atrophic gastritis and enteritis were associated with butyrate production and with an anti-inflammatory effect [20].

Firmicutes mostly comprise bacteria that can hydrolyze polypeptides and carbohydrates in the gut. Some Gram-negative Bacteroidetes in the gut encode many enzymes that degrade simple and complex polysaccharides in dietary fiber. Some Bacteroides members may play beneficial (often in the gut) and opportunistic pathogenic (in other body sites) roles based on their presence in the host. Bacteroidetes and Firmicutes have mutually reinforcing symbiotic relationships that promote energy absorption in the host. Some studies revealed an increased ratio of Firmicutes/Bacteroidetes in the gut compared with normal-weight individuals, proposing a biomarker for obesity, whereas the percentage was lower in diabetic patients [20]. In our current study, the Firmicutes and Bacteroidetes proportions were increased in *R. norvegicus*, indicating impaired intestinal integrity or infection in the body [20].

Urban rats can serve as reservoirs for some highly pathogenic microbes, such as *Bacillus anthraces*, *Yersinia pestis*, *Vibrio cholera*, etc. [5], that can lead to a severe endemic or pandemic. This study identified opportunistic and highly pathogenic bacteria in the urban rat stomach, gut, and lung samples. Pathogens such as *Corynebacterium pseudodiphtheriticum*, *Streptococcus parauberis*, *Rickettsiella agriotidis*, *Empedobacter brevis*, *Porphyromonas pogonae*, *Acetobacterium wieringae*, and *Rahnella aquatilis* were identified in lung samples. In contrast, *Corynebacterium pseudodiphtheriticum*, *Neisseria zaloph*, and Rhodococcus corynebacterioides were found in the stomach. We also found *Helicobacter muridarum* in a gut sample, an enterohepatic species linked to colitis and inflammatory bowel disease in humans. Other pathogens, such as *Macrococcus goetzii*, *L. johnsonii*, *Limosi-L. reuteri*, and *B. hinzii* (a small coccobacillus isolated from rodents that is reportedly involved in respiratory tract infections in birds and zoonotic transmission in immunocompromised people), *L. intestinalis*, *Helicobacter muridarum*, *Ralstonia insidiosa*, *E. coli*, *Limosi-L mucosae*, *C. perfringens*, *C. disporicum*, *C. paraputrificum*, *C. subterminale*, and *Pseudomonas mendocina*, were exclusively attributed to the stomach-, gut-, and lung-shared groups. The identification of these bacterial pathogens recommends a regular monitoring program for urban rodent distribution and the pathogens they carry and transmit to other mammals. Escherichia/Shigella belongs to the family Enterobacteriaceae and is one of the most common bacterial pathogens responsible for gut dysbiosis and dysentery [33]. A study reports increased Escherichia/Shigella proportions in mice suffering from gastric cancer [20]. In our study, *B. hinzii* was abundant in the lungs, gut, and stomach, indicating pulmonary and digestive infections in *R. norvegicus*. Moreover, several members of the Verrucomicrobiaceae reported in our study have previously been linked to glycosidase secretion for mucin degradation and intestinal integrity [34]. Verrucomicrobia from the family Verrucomicrobiaceae was detected in the *R. norvegicus* lung samples, suggesting its importance in promoting lung function.

The current study has several limitations. The small sample size made it difficult to develop animal models. In addition, we could not distinguish DNA from living or dead organisms, and the microbial differences between the lungs, stomach, and gut may be greater than the current estimate due to the movement of DNA from dead cells from one location to another [35]. However, the current findings may serve as a better baseline for future studies with larger sample sizes to confirm our findings. Metagenomics could help us better understand the role of microbiomes in *R. norvegicus* digestion and health status by comparing and contrasting microbial functions between distal gut regions. Detailed species-level data using metagenomics analysis would also make it possible to track the microbiota more effectively and solve problems related to over-splitting microbes due to the extreme variability of the 16S rRNA gene [36]. Moreover, larger-sample-size comparisons between free-living and captive *Rattus* species could also provide insights into the potential role of microbiomes within urban rat subspecies.

## 5. Conclusions

In conclusion, we identified several beneficial, opportunistic, and highly pathogenic bacteria in *R. norvegicus* stomach, gut, and lung samples, confirming the importance of the hidden inhabitants for rodents and other mammals’ health. Starting a routine survey program would be prudent to track and prevent the distribution of rodents and the pathogens they carry to humans and domestic animals.

## Figures and Tables

**Figure 1 microorganisms-11-02359-f001:**
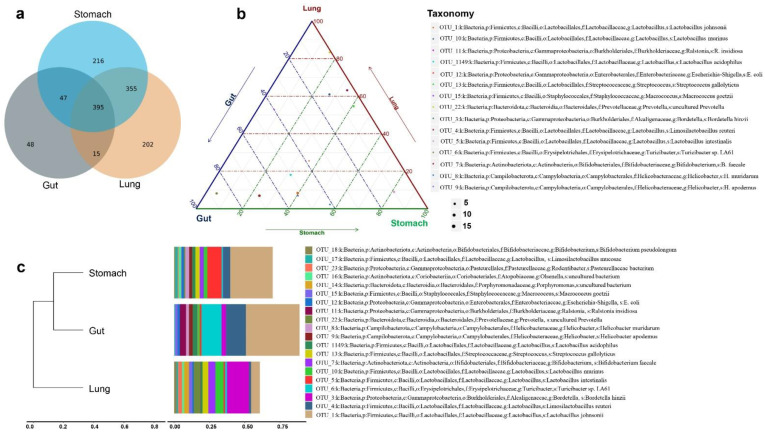
*R. norvegicus* stomach, gut, and lung microbial compositions. (**a**) The Venn diagram shows that 395 OTUs were shared by stomach, gut, and lung data; however, 216 OTUs are unique to the stomach, 48 to the gut, and 202 to the lung. (**b**) The details about the top 15 OUTs in the stomach, gut, and lung groups are shown. (**c**) Shows the distinct microbial communities between the three groups.

**Figure 2 microorganisms-11-02359-f002:**
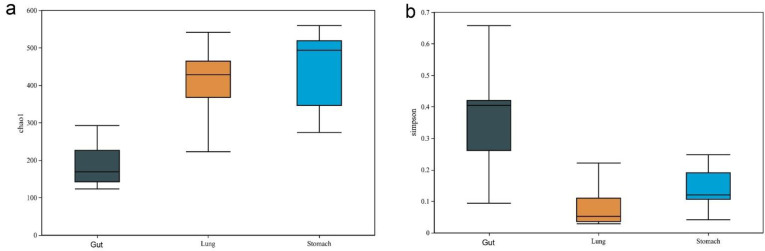
*R. norvegicus* stomach, gut, and lung microbial diversity. Alpha diversity quantifies differences in microbial composition at the species level. The (**a**) Chao1 (*p* = 0.009) and (**b**) Simpson (*p* = 0.02) diversity indexes showed significant differences between the stomach, gut, and lung groups.

**Figure 3 microorganisms-11-02359-f003:**
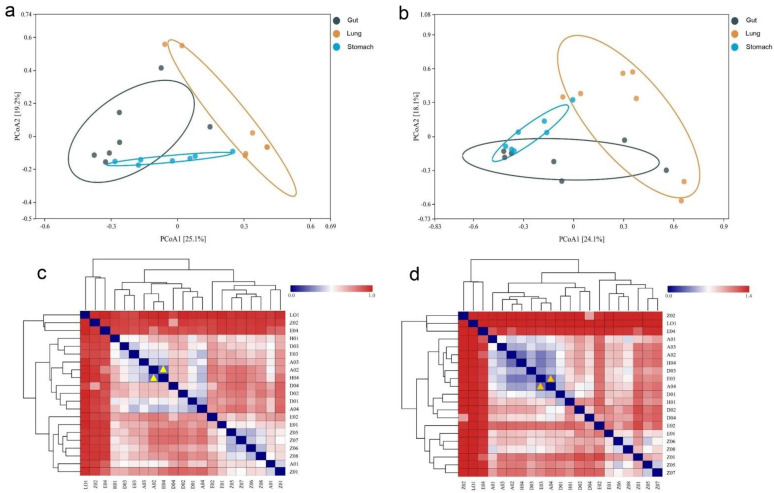
The Bray–Curtis and Euclidean distance metrics revealed significant differences among the microbial communities in the stomach, gut, and lung samples of *R. norvegicus*. (**a**) Using the Bray–Curtis distance, PCoA plots visualize the percentage variations between PCoA1 (25.1%) and PCoA2 (19.2%) among different samples. (**b**) Two PCoA coordinates based on Euclidean distances revealed percentage variations for PCoA1 (24.1%) and PCoA2 (18.1%) among the samples. (**c**) Bray–Curtis distances. The most similarity and lowest difference in the bacterial community profile was 0.2182 (shown with yellow triangles), observed between H04 and A02 (gut). (**d**) The lowest difference was 0.286 (shown with orange triangles), observed between A04 (gut) and E03 (stomach).

**Figure 4 microorganisms-11-02359-f004:**
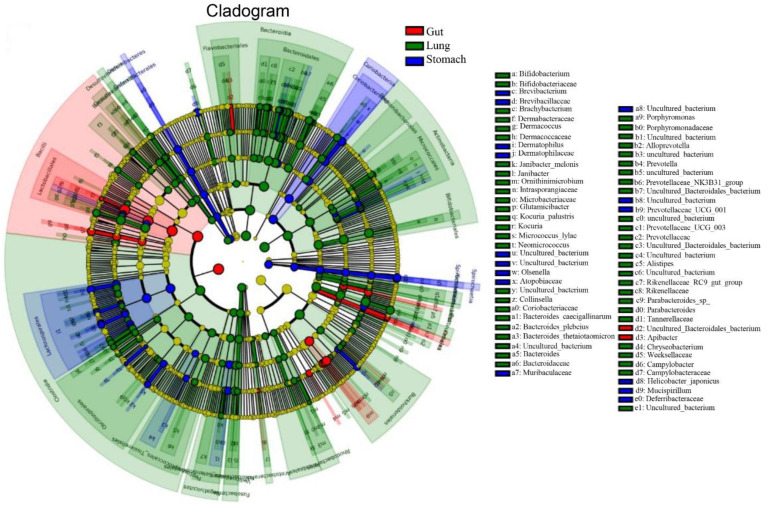
Overall, the cladogram shows 68 relatively abundant taxonomic clades (with an LDA score > 3.0) in the stomach (blue), gut (red), and lung (green) groups. Of the total, 15 clades existed in the stomach, 2 in the gut, and 51 in the lungs, consistent with the LEfSe result. In the lung, Burkholderiales were most abundant, followed by Actinobacteriota, Bacteroidota, etc.; in the stomach, Lachnospiraceae and Lachnospirales were abundant, while Apibacter and uncultured Bacteroidetes were abundant in the gut. The cladogram’s circles (each small circle) radiating from the inside to the outside represent different taxonomic groups. The diameters of the circles (each small) represent abundant bacterial taxa, and classifications with no discernible differences are highlighted in yellow.

**Figure 5 microorganisms-11-02359-f005:**
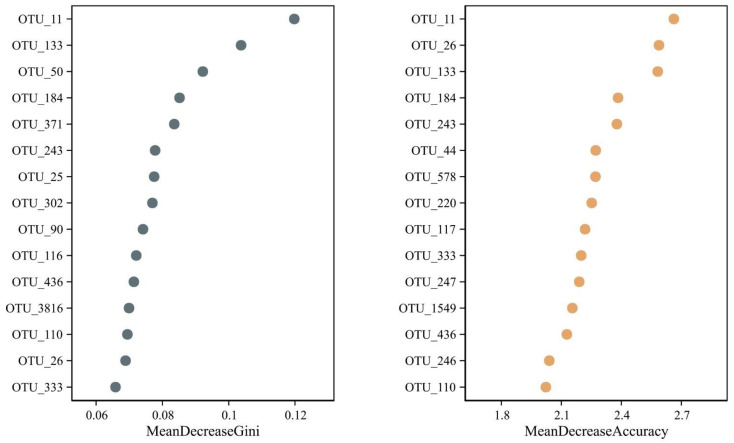
A random forest analysis determined the 15 most important OTUs with the highest discriminatory power between the three groups. Random forests of 10,000 trees were computed to generate the classifiers using the R package (v4.1.2). For each OUT, the mean decrease Gini and mean decrease Accuracy values were averaged, and the top 15 OTUs with the highest mean decrease Gini and mean decrease Accuracy values were plotted. OTUs with mean decreases in Gini mean decrease Accuracy values above the breakpoint curve were included in the classifier.

**Table 1 microorganisms-11-02359-t001:** Number of identified OTUs in stomach, gut, and lung groups.

Group	No. of OTUs	No. of Seq
Gut	301	99,055
Lung	804	85,523
Stomach	701	99,395

## Data Availability

Raw reads generated and analyzed during the current study were deposited in the NCBI Sequence Read Archive (SRA) under the accession number SRP449699.

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
