# Peer review of "A Comparative Analysis of the Stomach, Gut, and Lung Microbiomes in Rattus norvegicus"

_microorganisms, 2023, doi:10.3390/microorganisms11092359_

Round 1

Reviewer 1 Report

The manuscript describes the microbiota in different tissues, it is interesting and has good potential. However, it is necessary:

- Improve abstract

- There is a lack of information on how the phylogeny was carried out and how the data taken from the sequencing was organized, how the data was filtered, for example, which programs were used for each analysis.

- The results of the cladogram seem quite interesting to me, but it has been little explored. It lacked a better description of the findings, mainly the clades that are representatives only of gut, lung and stomach.

The english language minor editing. 

Author Response

Reviewer 1

Thank you for your insightful comments and suggestions on our manuscript's content. The revision was made in response to the reviewers' comments, and the changes are highlighted in red in the revised manuscript.

The comments, suggestions, and advice of reviewer 1 have been considered seriously as follows:

The manuscript describes the microbiota in different tissues, it is interesting and has good potential. However, it is necessary:

- Improve abstract

  1. There is a lack of information on how the phylogeny was carried out and how the data taken from the sequencing was organized, how the data was filtered, for example, which programs were used for each analysis.

Response: Thank you very much for the inside question; we added a brief statistical analysis in the abstract section (i.e., The USEARCH software assigned the dataset to operational taxonomic units (OTUs). The alpha diversity index was calculated using QIIME1, while the beta diversity index was determined using Bray-Curtis and Euclidean distances between groups. Principal component analyses visualize variation across samples based on the OTU information using the R package. Linear discriminant analysis, effect sizes (LEfSe), and phylogenetic investigation were used to identify differentially abundant taxa among groups..

  1. The results of the cladogram seem quite interesting to me, but it has been little explored. It lacked a better description of the findings, mainly the clades that are representatives only of gut, lung and stomach.

Response: Thank you very much for improving our article, we have added more information to the cladogram table 4 (i.e., Figure 4. Figure 4. Overall, the cladogram shows 68 relatively abundant taxonomic clades (with an LDA score >3.0) in the stomach (blue), gut (red), and lung (green) groups. Of the total, 15 clades existed in the stomach, two in the gut, and 51 in the lungs, consistent with the LEfSe result. In the lung, Burkholderiales were most abundant, followed by Actinobacteriota, Bacteroidota, etc.; in the stomach, Lachnospiraceae and Lachnospirales were abundant, while Apibacter and uncultured Bacteroidetes were abundant in the gut. The cladogram's circles (each small circle) radiating from the inside to the out-side represent different taxonomic groups. The diameter of the circles (each small) represents abundant bacterial taxa, and classifications with no discernible differences are highlighted in yellow.

  1. Comments on the Quality of English Language: The English language minor editing.

Response: The reviewer raised concerns about the language; therefore, we request that Syed Akbar Bangash, an Australian citizen, edit the final version of the manuscript. We hope the language is acceptable to the journal.

Reviewer 2 Report

The manuscript entitled “A comparative analysis of the stomach, gut, and lung microbiomes in Rattus norvegicus” is a concise and straightforward study for a modest set of samples. Based on the 16S rRNA metagenomics, the authors provide the microbial community structure of the gut, stomach, and lung of 7 healthy free-living R. norvegicus. Overall, the manuscript is well written and brings interesting information regarding the microbiome structure throughout the targeted parts of the digestive and respiratory tracts. After minor revisions the manuscript would be suitable for publishing.

Line 14: Please use (city or street rats) instead of wild rats. Or you can provide a short definition, something like this: “Rattus norvegicus is an ubiquitous urban pest…….”.

Line 31: remove “and”

Line 43: please use city or street rats instead of wild rats here and everywhere in the text.

Line 117-124: Provide a clear procedure for lib. construction. What the author would say by “  pooled before sequencing” ? the pool were performed by sample type or by animals? 

Line 249-262: Could the author provide more information regarding the threshold used to the species level identification (percent of identity/query cover). 

As the author used the V3–V4 region, they may limit the identification at the genus level since the V3–V4 region is not a discriminatory fragment at the species level.

Overall, The sentences are too lengthy, which causes confusion.

Author Response

Reviewer 2

Thank you for your insightful comments and suggestions on our manuscript's content. The revision was made in response to the reviewers' comments, and the changes are highlighted in red in the revised manuscript.

The comments, suggestions, and advice of reviewer 2 have been considered seriously as follows:

Comments and Suggestions for Authors

The manuscript entitled “A comparative analysis of the stomach, gut, and lung microbiomes in Rattus norvegicus” is a concise and straightforward study for a modest set of samples. Based on the 16S rRNA metagenomics, the authors provide the microbial community structure of the gut, stomach, and lung of 7 healthy free-living R. norvegicus. Overall, the manuscript is well written and brings interesting information regarding the microbiome structure throughout the targeted parts of the digestive and respiratory tracts. After minor revisions the manuscript would be suitable for publishing.

  1. Line 14: Please use (city or street rats) instead of wild rats. Or you can provide a short definition, something like this: "Rattus norvegicus is an ubiquitous urban pest…….".

Response: Thank you very much for the suggestion, Sir; we used the term "urban rats" instead of "wild rat" according to the reviewer's suggestion.

  1. Line 31: remove “and”

Response: We removed "and" from the keywords, as suggested by the reviewer.

  1. Line 43: please use city or street rats instead of wild rats here and everywhere in the text.

Response: Thank you very much, Sir; we have used term "urban rats" instead of "wild rat" throughout the manuscript.

  1. Line 117-124: Provide a clear procedure for lib. construction. What the author would say by “ pooled before sequencing” ? the pool were performed by sample type or by animals?

Response: Thank you very much, Sir; we have completely revised the bioinformatics analysis section in the revised manuscript, i.e., (The raw sequence data were cleaned by removing reads shorter than 230 bp, primer dimers, low-quality sequences (with ambiguous bases), and Q scores of ≥Q20 using the Illumina quality control toolkit. Primer dimers removed from the library reduce Illumina system specificity for sequencing the desired V4 region of the 16S rRNA gene. The Q30 sequencing score is a standard for high-quality sequencing with virtually no ambiguities or errors. Flash (v1.20) and Pear (v0.9.6) tools were used to merge clean reads into tags with a minimum 10 bp overlap and a 0.1 ‘unoise3’ algorithm from the USEARCH (v11) amplicon analysis tool. The SILVA database (v128) removed chimeras (sequences formed by incorrectly joining two or more sequences) in 16S rRNA data. All the clean sequences were clustered into operational taxonomic units (OTUs) using the USEARCH tool (v11) at 97% homology. Rarefaction curves were created to assess microbial richness based on the OUT data. This curve also shows how accurately a specific sample was sequenced to rep-resent its identity. The SILVA database (v128) was used to assign all sequences to different taxonomic groups.

The alpha diversity index (i.e., observed species, Chao1, Simpson, and phylogenetic distance tree) was calculated using QIIME1 (v1.9.1). The beta diversity index was evaluated using non-Euclidean (Bray-Curtis) and Euclidean distance dissimilarities between groups. Clustering and principal component analyses (PCA) were used to visualize variation across samples using the R package (v4.1.2).

  1. Line 249-262: Could the author provide more information regarding the threshold used to the species level identification (percent of identity/query cover).

Response: Thank you very much for the inside question, Sir, we received the 16S rRNA gene nucleotide data for each OUT (16S-based ID) from the sequencing data; we then individually subjected each sequence to the BLAST search, and "ezbiocloud.net/identify" search for species identification, based on the criteria as Top-hit taxon; Top-hit strain; Similarity(%); Top-hit taxonomy. Further, ezbiocloud is a beautiful tool for species identification based on 16S rRNA gene sequencing.

  1. As the author used the V3–V4 region, they may limit the identification at the genus level since the V3–V4 region is not a discriminatory fragment at the species level.

Response: Thank you very much for the inside question, Sir; we received the 16S rRNA gene nucleotide data for each OUT (16S-based ID) from the sequencing data. Although, some of the species were already identified during the initial analysis, Further, we individually subjected each sequence (16S-based ID) to the BLAST search and "ezbiocloud.net/identify" search for species identification, based on the criteria: Top-hit taxon; Top-hit strain; Similarity(%); Top-hit taxonomy. Further, "ezbiocloud" is a beautiful tool for species identification based on 16S rRNA gene sequencing.

  1. Comments on the Quality of English Language.

Response: The reviewer raised concerns about the language; therefore, we request that Syed Akbar Bangash, an Australian citizen, edit the final version of the manuscript. We hope the language is acceptable to the journal.

Round 2

Reviewer 1 Report

the authors significantly improved the manuscript, as well as modified the results to support the conclusions of the work.